# Hazara Orthonairovirus Nucleoprotein Antagonizes Type I Interferon Production by Inhibition of RIG-I Ubiquitination

**DOI:** 10.3390/v14091965

**Published:** 2022-09-04

**Authors:** Keisuke Ohta, Naoki Saka, Machiko Nishio

**Affiliations:** Department of Microbiology, School of Medicine, Wakayama Medical University, Wakayama, 811-1, Kimiidera, Wakayama 641-8509, Japan

**Keywords:** Hazara orthonairovirus, N protein, tripartite motif-containing protein 25, retinoic acid-inducible gene I, interferon

## Abstract

Viruses have evolved various strategies to evade the host innate immune system. The relationship between nairoviruses and the interferon (IFN) system is poorly understood. We investigated whether and how nairoviruses antagonize host innate immunity using Hazara orthonairovirus (HAZV) as a surrogate model for Crimean-Congo hemorrhagic fever virus. HAZV nucleoprotein (N) was found to interact with the tripartite motif-containing protein 25 (TRIM25). The N-terminal region of N protein and the C-terminal region of TRIM25 are important for their interaction. Overexpression of N protein results in weakened interaction of TRIM25 with retinoic acid-inducible gene I (RIG-I). Furthermore, K63-linked polyubiquitination of RIG-I is inhibited in the presence of N protein. Our data collectively suggest that HAZV N protein interferes with the binding of TRIM25 to RIG-I and subsequent K63-linked polyubiquitination of RIG-I, which leads to inhibition of type I IFN production.

## 1. Introduction

Hazara orthonairovirus (HAZV) belongs to the Orthonairovirus genus of the Nairoviridae family in the order Bunyavirales (http://talk.ictvvvonline.org/taxonomy) (accessed on 9 August 2022). It is closely related to Crimean-Congo hemorrhagic fever virus (CCHFV). CCHFV causes severe hemorrhagic fever with high lethality (approximately 30%), and is classified as a biosafety level (BSL) 4 agent due to its high pathogenicity. HAZV, by contrast, can be handled in BSL 2 because it is non-pathogenic to humans. HAZV is therefore a useful model for studying the virological characteristics of CCHFV. HAZV is an enveloped virus, and its genome consists of three segments of single-stranded, negative-sense RNA including S (small), M (medium), and L (large) genes [1]. S, M, and L genes encode nucleoprotein (N), glycoprotein (Gn and Gc), and RNA-dependent RNA polymerase (RdRp), respectively.

The innate immune system is the first line of host defense against virus infection. Innate immune cells recognize infection through pattern recognition receptors (PRRs), including retinoic acid-inducible gene I (RIG-I)-like receptors (RLRs) and Toll-like receptors (TLRs) [2,3]. RLRs are broadly expressed in most cell types, and can recognize infection of a variety of RNA viruses [4,5]. The RLR family consists of RIG-I, melanoma differentiation-associated factor 5 (MDA5), and laboratory of genetics and physiology 2 (LGP2). RIG-I and MDA5 contain two N-terminal caspase recruitment domains (CARDs), a central ATPase/helicase domain, and a C-terminal regulatory domain (RD), while LGP2 lacks CARDs [6]. Both RIG-I and MDA5 function as cytosolic receptors for viral RNAs, although they have different ligands. When ligand binding occurs, the CARDs of RIG-I and MDA5 undergo K63-linked polyubiquitination by E3 ubiquitin ligases. The tripartite motif-containing protein 25 (TRIM25) and RIPLET are E3 ubiquitin ligases for RIG-I ubiquitination, whereas TRIM65 is for MDA5 ubiquitination [7,8,9]. Ubiquitinated RIG-I and MDA5 then signal the adaptor, mitochondrial antiviral signaling (MAVS) protein, followed by activation of transcription factors including IFN response factor 3 (IRF3), IRF7, and nuclear factor κB (NFκB), leading to type I IFN production [10].

Viruses have evolved various mechanisms to antagonize the host IFN system. One means of inhibiting IFN production is by blocking RLRs and their related proteins. V proteins of paramyxoviruses, including human parainfluenza virus type 2 (hPIV-2), PIV-5, and mumps virus, bind to MDA5 and LGP2 to suppress the production of type I IFN [11,12]. Nonstructural protein 2 (NS2) of human orthopneumovirus (belonging to the Pneumoviridae family; formerly named human respiratory syncytial virus) binds to RIG-I to prevent association with MAVS [13]. TRIM25 was targeted by NS1 of influenza A virus (IAV) [14].

Nonstructural protein (NSs) of Rift Valley fever phlebovirus (RVFV) (belonging to the Phenuiviridae) has been first identified as an IFN antagonist among bunyavirus proteins, and its antagonism has been well characterized [15,16]. NSs of Dabie bandavirus (belonging to the same family as RVFV; formerly named ‘severe fever with thrombocytopenia syndrome virus’) also antagonizes IFN production [17,18]. Z proteins of New World arenaviruses prevent type I IFN production [19]. The ovarian tumor (OTU) domain of the L proteins encoded by some viruses in the Nairoviridae family including CCHFV and Dugbe orthonairovirus (DUGV) has the potential to suppress type I IFN production [20,21]. To date, the OTU domain is the only known putative antagonist of host IFN system in nairoviruses. Whether other viral proteins, including N and G proteins, are involved in IFN production remains unknown.

In this study, we investigated whether the HAZV N protein functions as an IFN antagonist. We demonstrate new strategies of how nairoviruses evade IFN production.

## 2. Materials and Methods

### 2.1. Cells and Virus

HeLa cells, COS cells, and SW13 cells were grown in Dulbecco’s modified Eagle’s minimal essential medium (DMEM) containing 5% fetal calf serum (FCS). HeLa cell lines constitutively expressing either wild type (wt) hPIV-2 V protein (HeLa/wt V) and its inactivated tryptophan (Trp) mutant (HeLa/V_W178H/W182E/W192A_) were previously described [22]. All cells were maintained in a humidified incubator at 37 °C with 5% CO_2_. In this study, we used HAZV (strain JC280) [23]. Sendai virus (SeV) strain Cantell was kindly provided by Dr. T. Sakaguchi (Hiroshima University, Hiroshima, Japan).

### 2.2. Antibodies and Reagents

Monoclonal antibody (mAb) against hPIV-2 V protein (315-1) and mAb against HAZV N protein (911-1) used for immunoblot have been previously described [24,25]. Anti-HAZV N mAbs (182-1, used for immunoblot; 3102-1, used for immunoprecipitation) were generated by immunization of BALB/c mice with HAZV-infected L929 cells. Hybridomas derived from splenocytes were cloned and cultured as previously described [25]. The mAb used for the detection of SeV N protein (828) has been previously described [26]. Anti-FLAG mAb and polyclonal Ab (pAb) were obtained from Sigma (St. Louis, MO, USA), anti-myc mAb from MBL (Nagoya, Japan). Anti-actin mAb was purchased from Wako (Osaka, Japan). Anti-STAT1 p84/p91 mAb and Anti-STAT2 pAb (C-20) were obtained from BD Transduction Laboratories (Lexington, KY, USA) and Santa Cruz Biotechnology (Santa Cruz, CA, USA), respectively. Anti-TRIM25 mAb was purchased from Abcam (Cambridge, MA, USA). Human IFN-α was obtained from Mochida Chemical Industries (Osaka, Japan).

### 2.3. Plasmids

HAZV N gene was cloned into the pcDNA3.1 vector (Invitrogen, Carlsbad, CA, USA) as previously described [25]. N gene with a FLAG tag at its N-terminal was also cloned into the pcDNA3.1 vector. Construction of N mutants was carried out by standard PCR mutagenesis methods. The cDNAs encoding RIG-I, MDA5, LGP2, TRIM25, or RIPLET with FLAG tag at their N-termini were inserted into pcDNA3.1 or pCI-neo vector (Promega, Madison, WI, USA). cDNAs of TRIM25 and its deletion mutants were also cloned into pEF4/Myc-His vector (Invitrogen). cDNAs of RIG-I and RIG-I-2CARDs were cloned into a pCMV-3Tag-8 vector with a 3x FLAG tag at their C-termini (Stratagene, La Jolla, CA, USA). Construction of ubiquitin mutant (Ub K63) with myc tag at its N-terminus was carried out by a standard PCR mutagenesis method, and cDNA of myc-Ub K63 was inserted into pEF4/Myc-His vector. These constructs were all confirmed by DNA sequencing.

### 2.4. Plaque Assay

SW13 cells grown in 12-well plates were infected with HAZV diluted serially 10-fold in DMEM without FCS and cultured in DMEM containing 2% FCS, 0.4% SeaKem ME agarose, and 0.4% SeaPlaque agarose (FMC Bioproducts, Rockland, ME, USA) until plaques were visible. The cells were then stained with 0.3% amido black.

### 2.5. Immunoblot and Immunoprecipitation Assays

Cells were harvested, and sonicated for 30 s three times in lysis buffer containing 50 mM Tris-HCl (pH7.4), 150 or 250 mM NaCl, 0.6% NP-40, and proteinase inhibitor cocktail (cOmplete, Roche, Basel, Switzerland), and centrifuged. Cell lysates were separated by SDS-PAGE, transferred to a nitrocellulose membrane, and analyzed by the western blot (WB) technique. For immunoprecipitation, the supernatants were incubated with Protein A Sepharose 4 Fast Flow (GE Healthcare Bio-Sciences, Piscataway, NJ, USA) preincubated with the appropriate Abs. Precipitated proteins were analyzed by WB.

### 2.6. RIG-I Ubiquitination Assay

COS cells were transfected with pEF4/Myc-His carrying myc-Ub K63 and pCMV-3Tag-8 carrying FLAG-tagged RIG-I or RIG-I-2CARDs, together with pcDNA carrying N protein using XtremeGENE HP (Roche) according to the manufacturer’s instructions. The cells were mock-infected or infected with SeV for 24 h, and were then lysed with lysis buffer containing 50 mM Tris-HCl (pH7.4), 50 mM NaCl, 0.6% NP-40, and cOmplete proteinase inhibitor cocktail. The cell lysates were immunoprecipitated with anti-FLAG pAb, followed by immunoblot.

### 2.7. Type I IFN Detection (Secreted Embryonic Alkaline Phosphatase (SEAP) Assay)

COS cells were transfected with pCMV-3Tag-8-RIG-I-2CARDs and pcDNA3.1-N using XtremeGENE HP. After two days, the supernatants were incubated with HEK-Blue IFN-α/β cells (InvivoGen, San Diego, CA, USA) overnight. The supernatants of the incubated HEK-Blue IFN-α/β cells were reacted with QUANTI-Blue solution for 1 h. SEAP activity was quantified by absorbance measurement at OD650 nm using microplate reader SH9000 (Corona Electric, Ibaraki, Japan). The concentrations of type I IFN were calculated from a standard curve made by serial dilutions of IFN-α.

## 3. Results

### 3.1. Type I IFN Inhibits the Growth of HAZV

We examined whether type I IFN affects HAZV growth. HeLa cells were stimulated with IFN-α for 24 h, followed by infection with HAZV at an MOI of 0.1 for 24 h. IFN-α treatment induced expression of STAT1 and STAT2 proteins (Figure 1A, lanes 2 and 4). Virus titers in the culture supernatant were determined by plaque assay. Based on the relative PFU/mL, HAZV growth was decreased by approximately 70% and 90% by IFN-α treatment at 24 and 48 h post-infection, respectively (Figure 1B).

We then used different HeLa cell lines (HeLa/wt V, which lacks STAT2 expression, or HeLa/V_W178H/W182E/W192A_, which does not affect STAT2 expression) (Figure 1C) [22]. HAZV growth in HeLa/wt V was approximately four-fold higher than that in its control cell line (HeLa/ctrl) (Figure 1D). In contrast, HAZV growth was not affected by the expression of hPIV-2 V_W178H/W182E/W192A_ (Figure 1D). These results indicate that HAZV growth is suppressed by IFN.

### 3.2. HAZV N Protein Binds to TRIM25

We hypothesized that HAZV encodes proteins that function to inhibit the type I interferon production pathway, similar to other viruses. hPIV-2 V protein has been reported to bind to MDA5 and LGP2 [11,12]. Using immunoprecipitation, we investigated whether HAZV N protein binds to IFN-related proteins. N protein, together with either FLAG-RIG-I, -LGP2, -MDA5, -TRIM25, or -RIPLET, was co-expressed in COS cells. Only TRIM25 was co-precipitated with N protein (Figure 2A, lanes 7 and 8), while none of the RLRs including RIG-I, LGP2, and MDA5, were co-precipitated with N protein (Figure 2A, lanes 1–6). N protein did not bind to RIPLET (Figure 2B, lane 2). To examine N-TRIM25 interaction in HAZV-infected cells, HeLa cells were infected with HAZV at an MOI of 1, and were subjected to immunoprecipitation. Endogenous TRIM25 was co-precipitated by anti-N mAb in HAZV-infected cells (Figure 2C, lane 2), indicating that TRIM25 interacts with N protein also in HAZV-infected cells.

N protein consists of three domains: N-terminal (aa 1–186), arm (aa 187–296) and C-terminal domains (aa 297–485) (Figure 3A) [27]. To determine the region of N protein important for binding with TRIM25, two deletion mutants of N protein (N ∆C and N ∆N) were prepared and subjected to immunoprecipitation with TRIM25 (Figure 3A). N ∆C (deletion of aa 187–485) retained the ability to bind to TRIM25, while N ∆N (deletion of aa 1–186) did not bind to TRIM25 (Figure 3B, lanes 2 and 4), indicating that the N-terminal region (aa 1–186) of N protein is important for the binding with TRIM25. TRIM25 consists of RING, two B-box (BB), coiled-coil (CCD), and SPRY domains (Figure 3C) [28]. We next identified the important region of TRIM25 using its deletion mutants (TRIM25 ∆C and ∆N) (Figure 3C). TRIM25 ∆N (deletion of aa 1–360) could still bind to N protein, although TRIM25 ∆C (deletion of aa 361–630) lost the binding capacity to N protein (Figure 3D, lanes 2 and 4), indicating that the C-terminal region (aa 361–630) of TRIM25 is necessary for binding with N protein.

### 3.3. N Protein Inhibits RIG-I Ubiquitination

To investigate whether N protein affects the TRIM25-RIG-I interaction, immunoprecipitation between TRIM25 and RIG-I was carried out in the presence of N protein. TRIM25-myc, FLAG-tagged RIG-I deletion mutant consisting of only the N-terminal two CARDs (RIG-I 2CARDs-FLAG; aa 1–614), and N protein were co-expressed in COS cells. TRIM25 was co-precipitated with RIG-I 2CARDs (Figure 4A, lane 2), indicating an interaction between TRIM25 and RIG-I. Co-expression of N protein resulted in a decrease in the amount of precipitated TRIM25 in a dose-dependent manner (Figure 4A, lanes 3 and 4, and Figure 4B). N protein was thus suggested to interfere with the interaction between RIG-I and TRIM25.

TRIM25 binding to RIG-I is necessary for RIG-I ubiquitination, so it was hypothesized that N protein inhibits TRIM25-mediated RIG-I ubiquitination. Expression of the RIG-I two CARDs alone induces K63-linked ubiquitination without any stimulation [7]. To examine the effects of N protein on RIG-I ubiquitination, myc-tagged ubiquitin mutant whose Lys residues, but not that of Lys63, were replaced with Arg (myc-Ub K63), RIG-I-2CARDs-FLAG, and N proteins were co-expressed in COS cells. RIG-I-2CARDs-FLAG was precipitated with anti-FLAG pAb, and ubiquitinated RIG-I was detected by myc mAb. Clear ubiquitination of RIG-I CARDs was detected (Figure 5A, lane 3). In contrast, the amount of ubiquitinated RIG-I was decreased to approximately 30% in the presence of N protein (Figure 5A, lane 4, and Figure 5B), indicating that N protein inhibits RIG-I ubiquitination. N ∆N, which does not bind to TRIM25 (Figure 3B), could not suppress RIG-I ubiquitination (Figure 5A, lane 5). The effects of N protein on SeV-induced RIG-I ubiquitination were also investigated. Full-length RIG-I (RIG-I-FLAG), myc-Ub K63, and N protein were expressed in COS cells, followed by SeV infection. Full-length RIG-I was ubiquitinated by SeV infection (Figure 5C, lane 3), consistent with the previous report by Gack et al. [7]. RIG-I ubiquitination was suppressed by exogenous expression of N protein, but not that of N ∆N (Figure 5C, lanes 4 and 5, and Figure 5D).

Subsequently, to investigate the effects of N protein on type I IFN induction, amounts of type I IFN in the culture supernatant were measured by SEAP assay. COS cells were transfected with RIG-I-2CARDs-FLAG and N protein. Two days post-transfection, the SEAP activity was measured, as described in the Materials and Methods section. Expression of RIG-I CARDs showed a robust increase in amounts of type I IFN in the culture supernatant (Figure 6). In contrast, N protein significantly suppressed RIG-I CARDs-mediated IFN response in a dose-dependent manner (Figure 6). IFN levels in cells expressing an increasing amount of N protein (N++) were approximately 6-fold lower than those in cells with N+ (Figure 6). N protein was clearly shown to inhibit RIG-I-mediated type I IFN induction.

## 4. Discussion

Viruses have evolved various mechanisms to evade the innate immune system. In the present study, HAZV N protein was found to suppress type I IFN production. Based on these results, we propose a model for antagonism of IFN production (Figure 7). TRIM25 binds to the first CARD of RIG-I at its C-terminal SPRY domain, which leads to the effective delivery of K63-linked polyubiquitin to the second CARD [7]. Activated RIG-I allows the recruitment of MAVS through RIG-I CARD and MAVS CARD interaction. Activated RIG-I/MAVS signaling causes phosphorylation and nuclear translocation of transcription factors including IRF3, leading to type I IFN production. The interaction of HAZV N protein with TRIM25 blocks the K63-linked ubiquitination of RIG-I, and the following RIG-I signal transduction (Figure 5). HAZV N protein binds to the C-terminal region (aa 361–630) of TRIM25 (Figure 3D). This region contains the SPRY domain, suggesting that interaction of RIG-I with TRIM25 is competitively inhibited by N protein (Figure 4). RIPLET rather than TRIM25 was recently reported to be the ubiquitin ligase for RIG-I [29,30,31]. Oshiumi et al. proposed a sequential ubiquitination model for RIG-I [32]. In this model, RIPLET first ubiquitinates the C-terminal region of RIG-I, which facilitates the second ubiquitination of RIG-I CARD by TRIM25. HAZV N protein did not bind to RIPLET or RIG-I itself but to TRIM25, which leads to suppression of RIG-I ubiquitination and subsequent RIG-I-mediated IFN response (Figure 2 and Figure 5). HAZV N protein might inhibit the second ubiquitination of RIG-I. N proteins of severe acute respiratory syndrome coronavirus (SARS-CoV), Middle East respiratory syndrome coronavirus, and SARS-CoV-2 also suppressed RIG-I ubiquitination by the same mechanism as HAZV N protein [33,34,35]. A similar pattern was reported in the 3B protein of foot-and-mouth disease virus, although this protein does not bind to TRIM25 but rather directly binds to RIG-I [36]. TRIM25 was also targeted by several viral proteins including NS1 of IAV [14], E6 oncoprotein of human papillomavirus [37], NSs of Dabie bandavirus [18], and the small T antigen (tAg) of polyomaviruses, such as JC virus and BK virus [38]. These proteins inactivate TRIM25 with unique and different strategies to suppress IFN production.

In the case of nairoviruses, the OTU domain of L protein has been focused upon as an IFN antagonist. The OTU domain of nairovirus L proteins, including CCHFV and DUGV, has protease activity for ubiquitin and ISG15 [20,21]. The OTU domain of HAZV L protein was reported to have at least deubiquitinating activity [39], indicating that HAZV has the ability both to inhibit ubiquitination and to remove ubiquitin. The isolated OTU domains have been reported to inhibit IFN induction [20,21]. However, the full-length L protein was found not to affect IFN response [20,40]. CCHFV cleaves off the 5′ triphosphate group of the genome to avoid RIG-I-dependent IFN induction [41], but with insufficient effect [42]. Nairovirus N protein might block the residual RIG-I-dependent IFN induction through inhibition of TRIM25.

Inhibition of the host innate immune system by bunyavirus proteins was first described in NSs of RVFV [17]. RVFV NSs suppress IFN production both by modulating host transcription and by blocking the nuclear export of host mRNA [18]. Viral NSs in other bunyaviruses, including La Crosse orthobunyavirus (Peribunyaviridae) and Dabie bandavirus (Phenuiviridae), were also found to act as IFN antagonists [17,43]. In contrast, to date, NSs of CCHFV (Nairoviridae) have not been reported to be an IFN antagonist, although it induces apoptotic cell death [44]. N proteins of nairoviruses would have more important roles in antagonism of the host IFN system than those of viruses belonging to other families.

In summary, we newly identified HAZV N protein as a repressor of IFN production. This study elucidated new strategies for how nairoviruses evade host IFN systems.

## Figures and Tables

**Figure 1 viruses-14-01965-f001:**
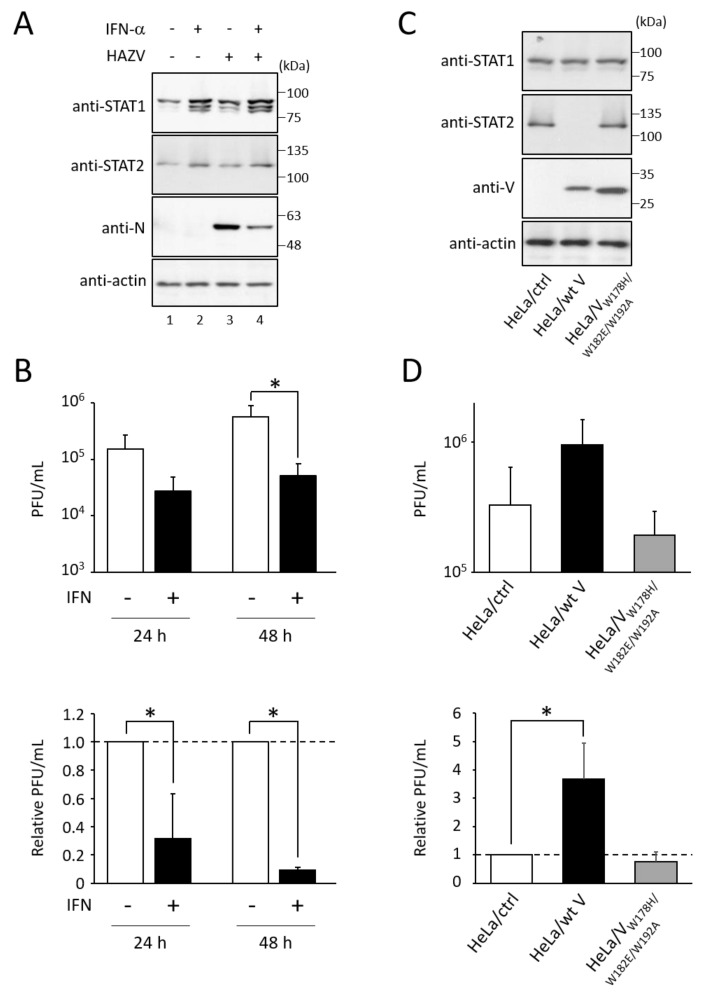
Effects of IFN on the growth of HAZV. (**A**) HeLa cells were stimulated with 1000 U/mL of IFN-α for 24 h, and were then infected with HAZV at an MOI of 0.1. After 24 h, cell lysates were subjected to immunoblot using the indicated Abs. Actin was used as a loading control. (**B**) The number of viruses in the culture supernatant of HAZV-infected cells after IFN-α treatment as in (**A**) was determined by plaque assay. The values of PFU/mL are the means from three independent experiments. Data are also shown as relative values (-IFN = 1). *p* values were calculated by the Student’s *t*-test. * *p* < 0.05, compared to values of -IFN. Error bars indicate standard deviations. (**C**) Lysates of HeLa cell lines constitutively expressing wt V of hPIV-2 (HeLa/wt V) or V_W178H/W182E/W192A_ (HeLa/V_W178H/W182E/W192A_), and their control cell line (HeLa/ctrl) were subjected to immunoblot using the indicated Abs. Actin was used as a loading control. (**D**) The cells were infected with HAZV at an MOI of 0.1 for 24 h, and virus titers were determined as described in (**B**). Data are shown as PFU/mL and relative PFU/mL (HeLa/ctrl = 1). *p* values were calculated by the Student’s *t*-test. * *p* < 0.05, compared to values of HeLa/ctrl.

**Figure 2 viruses-14-01965-f002:**
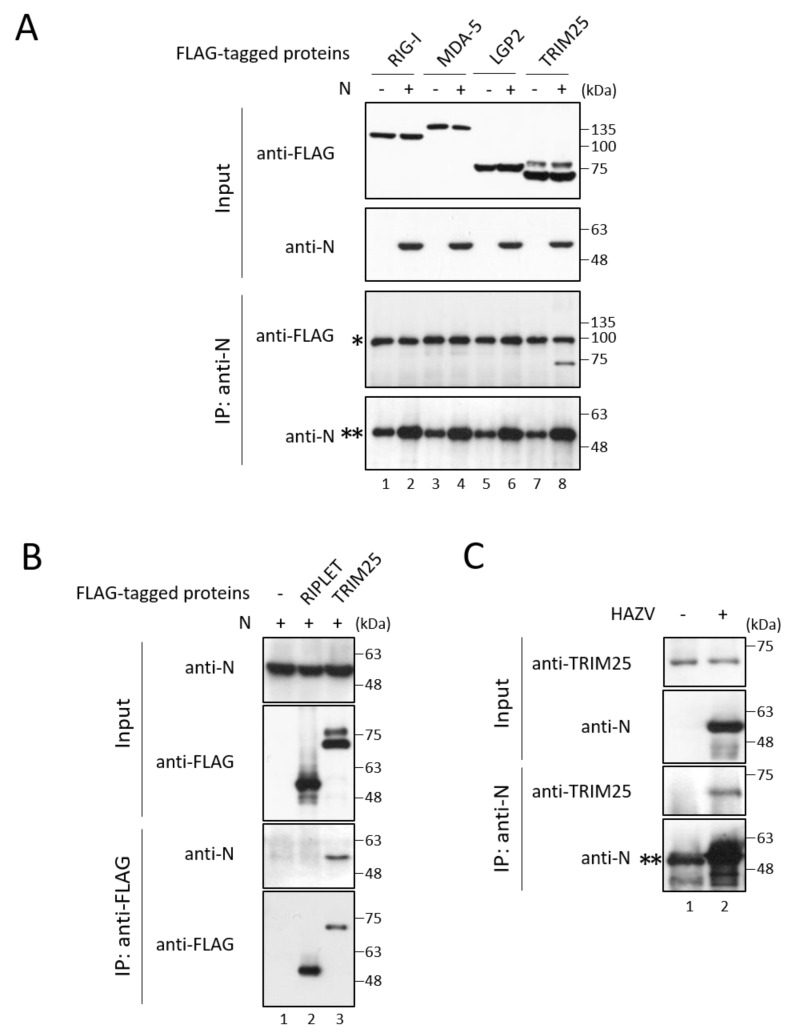
Interaction between N protein and IFN-related proteins. (**A**,**B**) COS cells were transfected with the indicated plasmids. After 48 h, cell lysates were directly analyzed by immunoblot (input). The cell lysates were immunoprecipitated with the indicated Ab, followed by immunoblot. A single asterisk indicates unknown non-specific bands. Double asterisks indicate an immunoglobulin heavy chain. (**C**) HeLa cells were either mock-infected or infected with HAZV at an MOI of 1.0. After 48 h, cell lysates were subjected to immunoblot and immunoprecipitation as described in (**A**). All experiments were performed at least three times independently.

**Figure 3 viruses-14-01965-f003:**
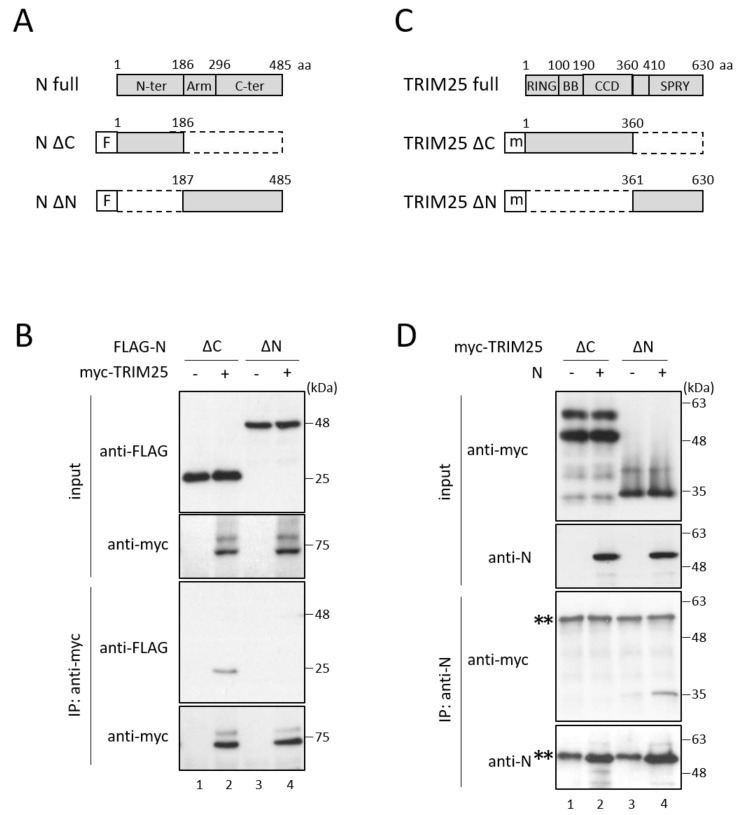
Identification of regions important for the interaction between N protein and TRIM25. (**A**,**C**) Schematic diagrams of (**A**) full-length N protein and its deletion mutants and (**C**) full-length TRIM25 and its deletion mutants. “F” and “m” indicate FLAG tag and myc tag, respectively. (**B**,**D**) COS cells were transfected with various combinations of the indicated plasmids. After 48 h, cell lysates were analyzed by immunoblot and immunoprecipitation assays (Figure 2). Double asterisks indicate immunoglobulin heavy chains.

**Figure 4 viruses-14-01965-f004:**
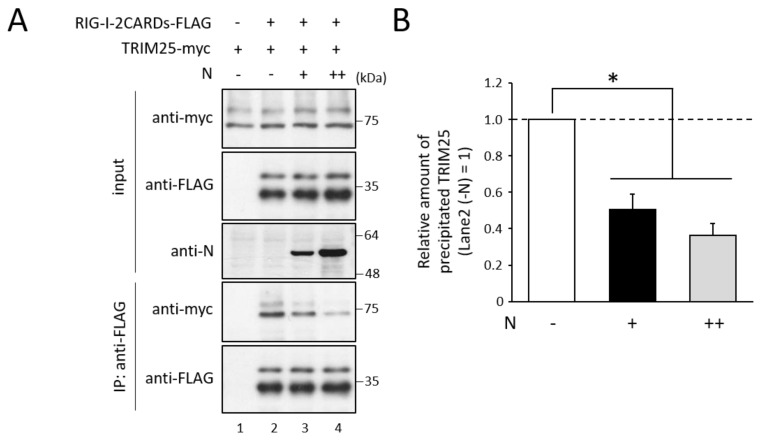
Effects of N protein on the interaction between RIG-I and TRIM25. (**A**) COS cells were transfected with the indicated plasmids. After 48 h, cell lysates were analyzed by immunoblot and immunoprecipitation assays (Figure 2). (**B**) The quantitative densitometry of precipitated TRIM25 in (A) was performed using ImageJ software (http://rsb.info.nih.gov/ij) (accessed on 22 January 2021). Data are the means from three independent experiments, and are shown as the relative value (lane 2 = 1). *p* values were calculated by the Student’s *t*-test. *, *p* < 0.05, compared to values of lane 2. Error bars indicate standard deviations.

**Figure 5 viruses-14-01965-f005:**
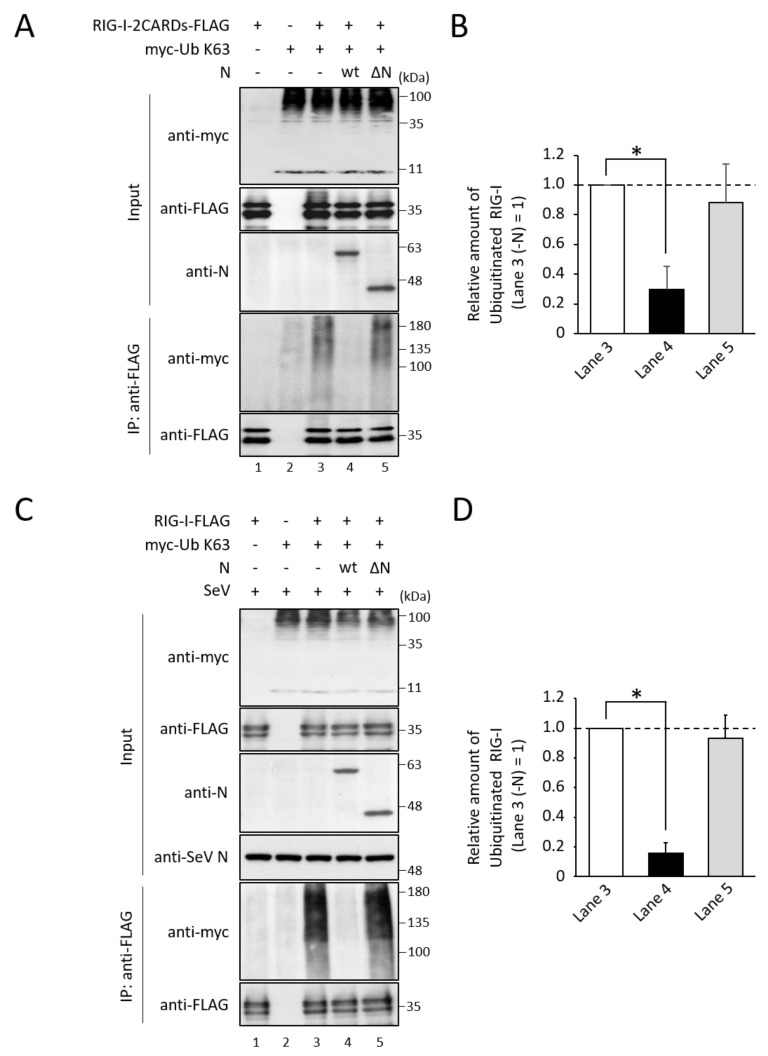
Effects of N protein on RIG-I ubiquitination. (**A**,**C**) RIG-I-2CARDs-FLAG (**A**) or full-length RIG-I (RIG-I-FLAG) (**C**), together with myc-Ub K63 and N protein was expressed in COS cells. After 24 h, cells were infected with SeV at an MOI of 10 for 24 h (**C**). Cell lysates were then analyzed by immunoblot and immunoprecipitation assays (Figure 2). (**B**) The quantitative densitometry of ubiquitinated RIG-I in (**A**,**C**) was performed as described in Figure 4B, and was shown in (**B**) and (**D**), respectively. Data are the means from three independent experiments, and are shown as the relative value (lane 3 (**A**) or lane 8 (**C**) = 1). *p* values were calculated by the Student’s *t* test. *, *p* < 0.05, compared to values of lane 3. Error bars indicate standard deviations.

**Figure 6 viruses-14-01965-f006:**
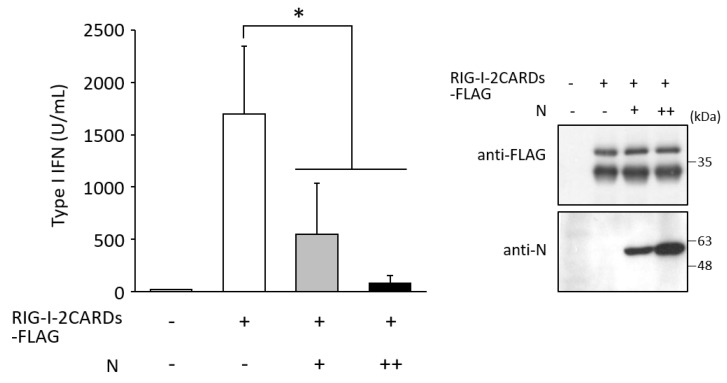
Effects of N protein on type I IFN production. COS cells were transfected with the indicated plasmids. ++ indicates twice the amount of pcDNA3.1-N as +. The supernatants were incubated with HEK-Blue IFN-α/β cells. The supernatants from HEK-Blue IFN-α/β cells were reacted with QUANTI-Blue solution, and SEAP activity was measured. SEAP activity was compared with a standard curve made by serial dilutions (10–10,000 U/mL) of IFN-α. Data are the means from four independent experiments. *p* values were calculated by the Student’s *t* test. *, *p* < 0.05. Error bars indicate standard deviations. Immunoblot shows the expression of RIG-I-2CARDs and N protein in transfected COS cells.

**Figure 7 viruses-14-01965-f007:**
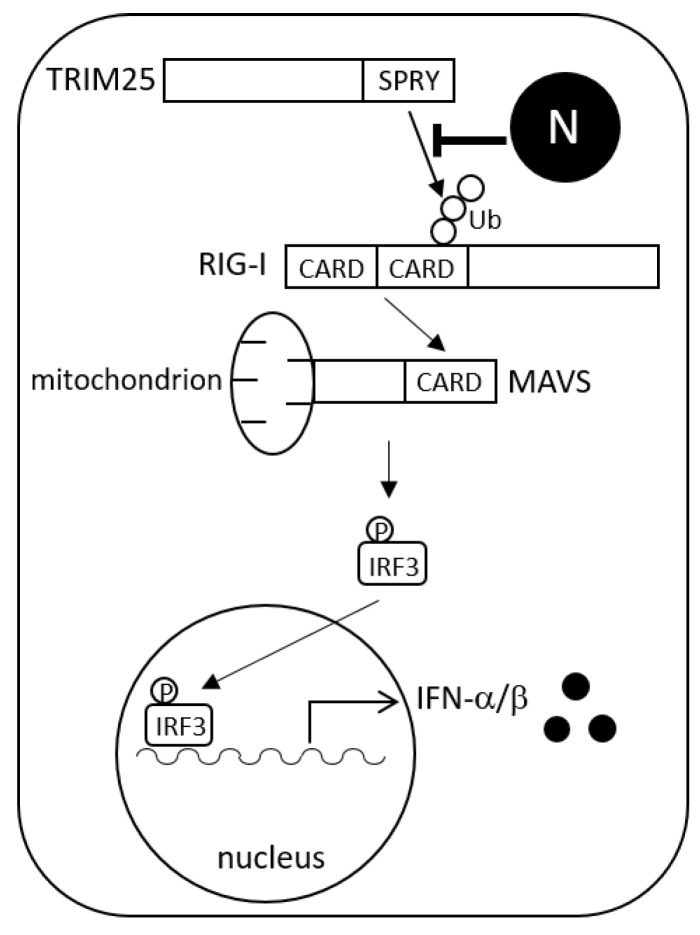
Models for inhibition of IFN production by HAZV N protein. N protein binds to TRIM25 to competitively inhibit the interaction between TRIM25 and RIG-I, resulting in the inhibition of K63-linked ubiquitination of RIG-I and subsequent type I IFN production.

## Data Availability

All datasets generated for this study are included within the article.

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
