# Peer review of "Hazara Orthonairovirus Nucleoprotein Antagonizes Type I Interferon Production by Inhibition of RIG-I Ubiquitination"

_viruses, 2022, doi:10.3390/v14091965_

Round 1

Reviewer 1 Report

The authors examined the effects of HAZV N for suppressing host innate immune functions.

General Comments

1. The experimental designs and approaches were straightforward. The obtained data seem to be consistent with the authors’ conclusions.

2. Fig. 1B and 1D.  The data (PFU/ml) shown in upper panels (24 h data in Fig. 1B and the control and wt V data in Fig. 1D) have no statistical differences, whereas the same data in lower panels, which are presented as relative PFU/ml, are statistically different.  It is puzzling why the same data shown in two different presentations gave different statistical significances. Although this reviewer is not an expert of statistical analysis, he/she considers that statistical significance should be the same in both panels.

3. Fig. 5C. To exclude the possibility that N expression suppressed SeV replication, leading to inhibition of ubiquitination of RIG-I, the authors should show accumulation of SeV protein or SeV RNA levels in Fig. 5C.

Specific comments

1. Lines 58-66. Inhibition of host innate immune responses by bunyavirus proteins was first described in NSs protein of Rift Valley fever virus, a phlebovirus in the order Bunyavirales. Also the mechanisms of the NSs-mediated suppression of innate immune responses have been well characterized. The authors should give a description of the mechanisms of the suppression of innate immune responses by NSs of Rift Valley fever virus.

2. Lines 71-75. They should include description of HeLa cells constitutively expressing hPIV-2 V and its variant, Trp mutant.

3. Lines 137-138. This sentence should be rewritten, as the description is not consistent with the data.

4. Line 140. Give full name of “Trp”.

Reviewer 2 Report

1. Line 245, Function of OTU domain.... is vague and sounds wrong. Please rephrase it. 

2. In Fig.1C the immunoblot for anti-V looks very saturated, a low exposure image will be more clear in showing the difference in size between wild type and mutant.

3. In Figure 6 legend or Methods there is no mention of what + and ++ sign indicates. Is ++ twice the amount of DNA as + or a different higher amount used for transfection ? Also a densitometry of the western blot may be added to show fold increase in N protein expression. It is also not clear if the  IFN levels decrease 5- fold or 50- fold in ++ condition as compared to +. These things should be mentioned in the Results and Legends. 

4. IP has been used to show all the interactions, can immunofluorescence images be shown to depict these interactions as a way to corroborate the extent of interactions?

5. In Figure 1A, 24h IFN treatment has been shown to decrease expression of N protein, does it further decrease after 48h? Please comment. 

6. Inf Figure 1A, the dose for viral infection is very low and IFN concentration used for stimulation is high. I am concerned if this effect is sustained at a higher MOI of HAZV infection?
